# The Effect of a Bioactive Oral System and CO_2_ Laser on Enamel Susceptibility to Acid Challenge

**DOI:** 10.3390/diagnostics13061087

**Published:** 2023-03-13

**Authors:** Mustafa Shubbar, Ali Addie, Lamis Al-Taee

**Affiliations:** 1Department of Conservative and Aesthetic Dentistry, Baghdad College of Dentistry, University of Baghdad, Baghdad P.O. Box 1417, Iraq; 2Center of Advanced Materials, Ministry of Science and Technology, Baghdad P.O. Box 0765, Iraq

**Keywords:** Regenerate Enamel Science, CO_2_ laser, pH-cycling, Raman microscopy, Vickers microhardness

## Abstract

This study evaluated the structural changes of enamel treated by the Regenerate system and carbon dioxide (CO_2_) laser against acid challenge. Thirty human enamel slabs were prepared and assigned into three groups: Group I: untreated (control); Group II: treated with the Regenerate system; and Group III exposed to CO_2_ laser. All specimens were subjected to an acid challenge (pH 4.5–7.0) for 14 days. Specimens were evaluated and compared at 120 points using five Raman microspectroscopic peaks; the phosphate vibrations ν1, ν2, ν3, and ν4 at 960, 433, 1029, and 579 cm^−1^, respectively, and the carbonate at 1070 cm^−1^, followed by Vickers microhardness test. The ratio of carbonate to phosphate was correlated to the equivalent microhardness numbers. The intensities of phosphate peaks ν1, ν2, and ν4 were reduced in all groups post-acid challenge, while the carbonate and ν3 were significantly increased (*p* < 0.000). Surfaces treated by Regenerate exhibited higher peak intensity of phosphate and carbonate before and after pH-cycling (*p* < 0.05). The mineral content in enamel had a direct effect on tissue microhardness, and the CO_2_-lased surfaces showed a reduced carbonate content and higher microhardness values. Both approaches induced surface changes that can protect enamel against acid challenge resulting in a significant benefit for dental healthcare.

## 1. Introduction

Tooth enamel is subjected to demineralization on exposure to acids, followed by remineralization by the effect of saliva. The net loss or gain of minerals over time will lead to carious progression or to being stabilized and regressed. This depends on optimizing the caries-preventive measures to reinforce tooth structure against decay and encouraging tooth remineralization [1]. Fluoride is the most common therapeutic agent that has been used to protect the enamel from the acid challenge and support tissue repair [2]. To improve the benefits of fluoride-contained dentifrices, calcium-contained products were introduced in an attempt to increase the availability of calcium ions in plaque and saliva and enhance the retention of fluoride in the oral cavity [3], which, in turn, encourages tissue remineralization [4]. Moreover, the use of bioactive materials supplied by various delivery vehicles is proposed to integrate the hydroxyapatite nanoparticles forming a biomimetic firm layer that preserve enamel structure and morphology [5]. One of these systems is the Regenerate Enamel Science which consists of a toothpaste containing calcium silicate and sodium phosphate salts combined with a dual-phase gel system composed of calcium silicate/sodium phosphate salts in one phase and sodium fluoride in the other phase [6,7]. Previous studies [8,9] supported the repair and protective action by the deposition of calcium silicate on acid-eroded enamel surfaces. They assumed that the calcium silicate could be transformed into hydroxyapatite and then deposited on both intact and eroded enamel surfaces, providing significant protection against erosive challenges. However, Tomaz et al., 2020 [10] found that this protection was limited to the superficial layer of enamel only. Ideally, the remineralizing systems must provide calcium, phosphate, and fluoride ions to allow mineral deposition at the subsurface layer of enamel; however, the regenerative effect of fluoride and calcium-contained toothpaste was restricted to about 30 μm depth into the enamel surface [11], with weak mechanical properties of these surfaces in comparison to intact enamel. Nevertheless, the formation of less-soluble surface hydroxyapatite is advantageous to increase enamel resistance to acid challenges [8]. This necessitates further investigation of the effect of these products on the structural/ morphological and mechanical properties of enamel surfaces when subjected to an acid-challenging environment.

The carbon dioxide (CO_2_) laser is another therapeutic approach that can reduce the solubility of enamel to acid dissolution and decrease the carbonate content, which will enhance the resistance to dental caries. It induces structural changes within enamel via the melting and re-crystallization of hydroxyapatite crystals, thereby enhancing the resistance of eroded enamel to demineralization [12]. The application of CO_2_ laser at 10.6 μm wavelength is highly recommended since it penetrates ten times deeper without damaging the surface nor causing an increase in the pulp temperature [13,14]. Moreover, the absorption of this wavelength is close to the phosphate, carbonate, and hydroxyl groups in the hydroxyapatite [15]. Once the light is absorbed by minerals, structural and chemical alterations in enamel crystals will occur, manifested by thermal decomposition of the carbonated apatite, fusion, and recrystallization of hydroxyapatite crystals [16,17]. These surface changes occur within the depth of 58 μm, which is sufficient to reduce the demineralization up to 98% [18]. However, the use of high power and energy of CO_2_ laser with inappropriate parameters can lead to cracks and irregularities in the tooth surface, increasing the brittleness and reducing the enamel hardness, which serves as points for the initiation of acid attacks [19]. Nevertheless, if appropriate parameters are selected, they can induce favorable changes within the enamel structure and thus increasing its resistance to acid attack and demineralization.

Confocal Raman Microspectroscopy is a valuable tool to analyze the molecular structure of biological tissues and detect structural alterations through their specific molecular vibrational energy signatures. It can quantitatively measure the mineral distribution within dental hard tissues, which will help assess the degree of demineralization or remineralization [20]. The non-invasive nature of this technique enables correlating the mineral contents to the mechanical properties of the surface by using the same areas. Although the mineral distribution was previously investigated, [21] no previous study was conducted to assess the structural alteration in enamel following various surface treatments. 

Surface microhardness is also considered a direct measure of the mineral content in dental hard tissues and an indirect indicator for mineral gain or loss. This means that the higher the tissue hardness, the greater the surface resistance to acid attack [22]. Accordingly, the present study evaluated and compared the structural changes in tooth enamel treated by the Regenerate Enamel Science system and carbon dioxide (CO_2_) laser before and after pH-cycling using Raman microspectroscopy and Vickers microhardness. In an attempt to find a correlation between the chemical and mechanical properties of enamel, the study assessed if there was a statistical correlation between the carbonate to phosphate Raman peak ratio (1070/960 cm^−1^) and their corresponding microhardness number at specific points in the treated and untreated surfaces before and after acid challenge. The null hypothesis: (i) there were statistically no significant differences in Raman mineral peaks intensities and VHN between the experimental and control groups before and after pH-cycling; (ii) there was no statistical correlation between Raman peak ratio of carbonate to phosphate (1070/960 cm^−1^) and the equivalent VHN within each group.

## 2. Materials and Methods

### 2.1. Specimens Preparation

Ten permanent, caries-free human premolars were collected from patients < 20 yrs for orthodontic treatment using an ethics protocol approved by the health research committee (Ref No. 471522, 19/1/2022), then stored in de-ionized water in a cold cabinet (+4 °C). The roots were sectioned at the Cemento-Enamel Junction (CEJ) (Isomet 1000, Buehler, Lake Bluff, IL, USA) using a water-cooled diamond blade (330-CA/RS-70300, Struers, Detroit Rd. Westlake, LLC, Cleveland, OH, USA). Each crown was hemi-sectioned mesiodistally into two halves, and only the buccal half was obtained. Each surface was further divided into 3 slabs (4.0 × 4.0 × 2.0 mm) and embedded in epoxy resin molds. All specimens were polished (Laryee Technology CO.LTD, China) under water cooling using silicon carbide paper in a sequential pattern P1200 for 10 s, P2500 for 10 s and P4000 for 4 min to obtain smooth and flat surfaces, followed by ultrasonic cleaning for 4 min to remove surface debris [23]. The slabs (n = 30) were assigned to three experimental groups (n = 10 per group): Group I: sound enamel surface did not receive any treatment (control); Group II: enamel surface received a frequent application of Regenerate Enamel Science System (serum/paste); and Group III: enamel surface subjected to CO_2_ Laser. Then all groups were subjected to a pH-cycling (4.5–7) for 14 days. 

### 2.2. The Application of Regenerate Enamel Science System

Regenerate Enamel Science System is composed of a regenerating serum and activator combined with toothpaste. The serum and the toothpaste contain the same ingredients (Glycerin, Calcium Silicate, PEG-8, Trisodium Phosphate, Sodium Phosphate, Aqua, PEG-60, Sodium Lauryl Sulfate, Sodium Monofluorophosphate, Aroma, Hydrated Silica, Synthetic Fluorphlogopite, Sodium Saccharin, Polyacrylic Acid, Tin Oxide, and Limonene, while the activator gel contains Aqua, Glycerin, Cellulose Gum, Sodium Fluoride, Benzyl Alcohol, Ethylhexylglycerin, Phenoxyethanol, and Sodium Fluoride (1450 ppm F). In Group II, the Regenerate Enamel Science System (gel/paste) was applied to the enamel surface following the manufacture instructions, in which the serum (serum and activator) was mixed by magnetic stir and then applied to the enamel surface for three minutes using a disposable brush for three consecutive days. Then freshly prepared slurries of toothpaste (4 mL per specimen) were mixed with distilled water for 4 min using a magnetic stir until homogenous using a 1:3 dilution ratio, then applied for 2 min on each slab for 13 days [4]. All procedures were carried out by the same researcher following the manufacturers’ instructions. These specimens were stored in de-ionized water until being analyzed by Raman microspectroscopy, SEM-EDX, and then microhardness.

### 2.3. The Application of Carbon Dioxide Laser CO_2_

A commercially pulsed CO_2_ laser (CO_2_ Fractional Laser, JHC1180, China) was applied to enamel surfaces in Group III using the following parameters: 2 W power; 10 ms pulse duration; 50 Hz pulse frequency; 0.2 mm focal spot; and 11.5 J/cm^2^ energy. Surface scanning was performed for 5 s from an X–Y positioning platform in noncontact mode keeping a 10 mm distance between the tip of a handpiece and the surface accompanied by water cooling to mimic the clinical conditions in preserving the vitality of dental pulp [24].

### 2.4. Acid Challenge (pH-Cycling)

The pH-cycling process was performed following the Featherstone model, which was later modified by Amaechi et al. (2019) [25,26]. The demineralizing and remineralizing solutions were prepared and refreshed every 5 cycles. Firstly, specimens were kept in the remineralizing solution for 24 h (37 °C), followed by six h in a demineralizing solution (40 mL per sample) that contains 2.0 mmol/L of Calcium, 0.472 g/L of Ca (NO3)_2_-4H_2_O, 2.0 mmol/L of Phosphate, in a buffer (pH = 4.5) contains 75.0 mmol/L Acetic acid, 0.272 g/L of KH_2_PO_4_, and 4.508 g/L of CH_3_COOH. Then they were kept in a remineralizing solution (20 mL per specimen) overnight. It contains 1.5 mmol/L of Calcium, 0.354 g/L of Ca (NO3)_2_-4H_2_O, 0.9 mmol/L of Phosphate, 0.1225 g/L KH_2_PO_4_, 130 mmol/L of KCI, 9.691 g/L KCl, 2.0 mmol/L NaCacodylate, and 4.28 g/L NaC2H_6_AsO_2_-3H_2_O, pH = 7.0). The pH-cycling process was repeated for up to 14 days. After that, the specimens remained in the remineralizing solution for two days at 37 °C and then washed thoroughly for 10 s and stored in distilled water until further assessment.

### 2.5. Raman Microspectroscopy

The 120-point scans were made over 30 enamel specimens before and after pH-cycling (n = 4 per sample) using a high-resolution confocal laser Raman microspectroscopy (Senterra, Bruker Optics, Germany) operating in line scan mode with a 780 nm near-infrared diode laser and a grating of 400 lines/mm. Spectra were measured over the range of 80–3800 cm^−1^ using a laser power of 100 mW with an integration time of 30 s at each point. The scanned points were positioned 500 μm from the top and bottom edges at the center of the enamel slab before pH-cycling (Figure 1). After pH-cycling, the examined points were positioned 800 μm from previously scanned points. This was achieved using a programable stage with a 1μm resolution area. After acquisition and spectra processing (baseline correction) using Raman software (OPUS, Bruker Optics, Germany), five Raman spectroscopic peaks were identified. The intensity of the phosphate peaks ν_1_, ν_2_, ν_3_, and ν_4_ vibration was at 960, 433, 1029, and 579 cm^−1^, respectively, and the intensity of the carbonate peak at 1070 cm^−1^. The peak intensities were calculated before and after the pH-cycle and averaged.

### 2.6. Scanning Electron Microscopy-Energy Dispersive Spectroscopy (SEM-EDX) Analysis

Three representative specimens from each group (before and after pH-cycling) were dried and carbon-coated in a vacuum chamber (YKY SEM Coating System). The samples were examined under Scanning Electron Microscopy coupled to an energy-dispersive X-ray spectroscope (INSPECT F50, FEI Company, Eindhoven, The Netherlands) with an accelerating voltage of 30 kV, using two magnification powers of 1000× and 6000×, and working distances of 100 µm and 10 µm, respectively.

### 2.7. Vickers Microhardness

The Hardness profile was examined by a Vickers microhardness tester (TH715, Obsnap Instruments Sdn Bhd, Selangor, Malaysia) using a diamond square-based pyramid diamond-shaped indenter with a 200 gf load for 15 s [4]. A total of 120 indentations (n = 4 per sample) were made on the same points that were previously assessed by Raman microspectroscopy before and after pH-cycling (Figure 1). The Vickers hardness number was recorded automatically using the manufacturer’s software.

### 2.8. Statistical Analyses

The statistical analysis was performed using SPSS software version 26 (IBM, Chicago, IL, USA). The Shapiro–Wilk test was used to evaluate the normality of data distribution. The data was statistically analyzed using One-way ANOVA followed by Tukey post-hoc multiple comparisons regarding the intensities of Raman peaks (A.U.) and Vickers microhardness number (VHN). The independent *t*-test (Minitab 14, Minitab LLC, Chicago, IL, USA) assessed the differences before and after pH- cycling. The probability level for statistical significance was set at α = 0.05. Pearson’s correlation coefficient test was used to explore if there was a correlation between Raman phosphate: carbonate peak ratio and their equivalent Vickers hardness number (VHN) at each point. The peak ratios were determined by dividing the intensity of phosphate ν_1_ to carbonate (960 cm^−1^/1070 cm^−1^).

## 3. Results

### 3.1. Chemical Analysis of Enamel before and after pH-Cycling

The relative Raman band intensities (Mean ± SD) with statistical correlations of all groups before and after pH-cycling are shown in Table 1. All enamel surfaces exhibited the presence of four characteristic phosphate peaks, ν_1_ at 960 cm^−1^ (symmetric P-O stretching vibrational mode), which is the strongest peak spectrum, ν_2_ at 433 cm^−1^ (symmetric bending vibrational mode), ν_3_ at 1029 cm^−1^ (asymmetric stretching vibrational mode), and ν_4_ at 579 cm^−1^ (asymmetric bending vibrational mode). Additionally, the carbonate peak intensity was observed at 1070 cm^−1^, which is the main substituent in the crystalline structure of the biological hydroxyapatite (HAp). The intensities of phosphate peaks (ν_1_, ν_2_, ν_4_) were significantly reduced in all groups after pH-cycling (independent *t*-test, *p* < 0.000), except that of ν_3_ peak at 1029 cm^−1^, which was significantly increased after exposure to acid challenge. The carbonate peak intensity at 1070 cm^−1^ was increased three times in both Regenerate (group II) and the control (group I) in comparison to their values before pH-cycling (Independent *t*-test, *p* < 0.05).

One-way ANOVA analysis showed statistically significant differences between groups regarding Raman peak intensities (*p* = 0.000) before and after pH-cycling. Before pH-cycling, Group II (surfaces treated by the Regenerate system) recorded the highest intensities of phosphate and carbonate (*p* > 0.000) among the groups. The surfaces exposed to CO_2_ (Group III) showed higher intensities of phosphate peaks (ν_1_, ν_2,_ and ν_4_) than the control, but the peak intensities of carbonate and ν_3_ peak intensities were statistically significantly the lowest among groups (*p* = 0.000), Table 1, (Figure 2). After pH-cycling, all Raman phosphate peak intensities (ν_1_, ν_2,_ and ν_4_) were statistically significantly reduced in comparison to those before the acid challenge. Nevertheless, Group II (Regenerate Enamel system) exhibited the highest mean intensity values of all Raman peaks among groups (*p* < 0.05). Group III (CO_2_ treated surfaces) showed a similar result to that before the pH-cycling, as the carbonate and ν_3_ peak intensities showed the lowest values than other groups (*p* = 0.000), (Table 1), (Figure 3). 

### 3.2. Vickers Microhardness

All enamel surfaces showed a statistically significant reduction in their hardness values after pH-cycling as compared to their values before pH-cycling (*p* < 0.000, Independent *t*-test). The greater difference in values was observed in the untreated surfaces (group I) that decreased threefold after exposure to the acid challenge. The CO_2_ laser-treated surfaces (group III) recorded statistically highest VHN (*p* < 0.000) before and after pH-cycling, followed by those treated by the Regenerate system (*p* < 0.000), while the control group recoded the lowest mean value among all groups (*p* < 0.000), (Figure 4). 

## 4. Discussion

The optimal therapeutic approach that protects tooth enamel and enhances the resistance to dental caries is always recommended but remains challenging due to the structural and functional complexities. This paper investigated the effectiveness of using the Regenerate Enamel Science System and CO_2_ laser to protect enamel against acid challenges. The results supported the beneficial effect of both systems in changing the enamel surface and enhancing the hardness property, with the ability of Regenerate to reintegrate to the enamel surface, forming a biomimetic film that preserved the structure and morphology of enamel after exposure to acids. Accordingly, the first stated hypothesis was rejected.

Raman microspectroscopy is a highly selective technique used for probing the molecular species in mineralized dental tissues and can quantitatively measure the demineralization in enamel with a sensitivity of 97% and specificity of 100% [27]. Raman phosphate peaks ν_1_, ν_2_, ν_3_, and ν_4_ were observed at the same positions (960, 433, 1029, and 579 cm^−1^, respectively) as previously determined in the literature [28,29]. These peaks indicate the presence of phosphate-based crystalline minerals in enamel; however, ν_2_, ν_3_, and ν_4_ showed weak spectral intensity compared to ν_1_ at 960 cm^−1,^ which was the strongest signal among all Raman spectra (Figure 2 and Figure 3) and which correlated to the degree of demineralization in tooth tissues [30]. 

There was a considerable reduction in the intensity of phosphate peaks in all groups after pH-cycling (*p* < 0.000, Table 1), which might indicate the dissolution of the hydroxyapatite (HAp) crystals in enamel when subjected to an acidic condition [27,31]. However, the ν_3_ phosphate peak intensity was increased after exposure to acid, due to an overlap that might occur with the carbonate spectra in the range between 1000–1079 cm^−1^, which also showed an increase in its intensity after pH-cycling (Figure 2 and Figure 3). This reduction in peak intensities was combined with a decrease in the atomic concentrations of calcium and phosphorous ions in all groups post-pH-cycling, which means that these ions are either dissolute and washed away, or consumed through the formation of the unstable compound that released from surfaces which might necessitate further investigation. During immersion in the demineralizing solution, the hydrogen ions diffuse into the enamel while the calcium and phosphate ions are released, mimicking the dynamic process of the carious process associated with the dissolution of the hydroxyapatite crystals. The first affected areas are the inter-crystalline and inter-prismatic spaces in the enamel, which promotes the development of defects in holes and cracks in the enamel surfaces. In lased surfaces, some of these ions might be trapped in the micro-spaces instead of being diffused into the surrounding solution. Although laser irradiation may fail to induce remineralization, it may inhibit demineralization with a significant enhancement in tissue microhardness (Figure 4), thus preventing the penetration of acids and reducing enamel dissolution [31].

Enamel surfaces treated by Regenerate exhibited a remarkable increase in all Raman peak intensities before and after pH-cycling (*p* < 0.000), Table 1. This might be attributed to the deposition of calcium and phosphate in the form of a mineral-like structure coating the enamel surface, as shown in Figure 5 (RB, RA). The Regenerate system is composed of calcium silicate and phosphate, in which the calcium ions can be exchanged with the H^+^ in the surrounding fluid, resulting in the formation of a silanol compound (Si-OH) at the enamel surface. The increase in pH creates a negatively charged surface, on which the presence of Si-O forms a template that supports the precipitating of minerals [8,31]. These deposits can become active during the repair process against the acid challenge process, making the enamel more resistant to acid dissolution [31]. The higher atomic number of Ca and P concentrations at the enamel surface (Figure 6, RB, RA) might support these speculations, added to the presence of Si ions that remained attached to the surface even after the pH-cycling. This finding is supported by a clinical in situ study that confirmed the deposition of new minerals on a bovine insert after 4 weeks of brushing with a toothpaste containing calcium silicate and sodium phosphate salts [31]. Moreover, the presence of calcium salts and phosphate might strengthen the fluoride release, which enhances the re-hardening capacity of enamel when exposed to the acid attack in comparison to conventional fluoride dentifrices [5,32] and thereby reducing the demineralization process and promoting enamel remineralization. This justified the higher microhardness values of this group that appeared less affected by the acidic change under SEM (Figure 5, RB, and RA) in comparison to the control that showed rough surface with furrows, grooves, and scratches associated with greater prism exposure (Figure 5, CB, and CA). This is consistent with Esfahani et al. (2015) [33], who reported an enhanced microhardness of demineralized enamel following the application of Remin Pro, which contains calcium and phosphate in the form of hydroxyapatite that might precipitate and fill the exposed enamel lesions.

The carbonate (CO_3_^2-^) is another key element in tooth enamel which makes up 2–5% of its weight. It is a fundamental substituent in the crystalline structure of the biological hydroxyapatite (HAp) that influences the enamel susceptibility to caries. It was assessed by Raman at an absorption band of 1070 cm^-1^, which was the lowest in CO_2_-treated surfaces before and after the acid challenge (*p* < 0.000). This referred to the thermal effect of CO_2_ laser that reduces water and carbonate contents in enamel [34] and, thus, reduces the solubility of enamel and enhances the resistance to acid attack. It has been suggested that the carbonate could be adsorbed or trapped in the form of amorphous calcium carbonate or replace the phosphate in the fused apatite. However, no evidence is proven yet for the substitution site of the carbonate in the crystal lattice [35]. This is supported by Kuramochi et al. (2016) [21], who found that the carbonate ions incorporated into remineralized enamel lesions, which suggests that the carbonate will not suppress the incorporation of phosphate into lesion nor inhibit tissue remineralization.

On the other hand, the intensities of phosphate peaks were higher in CO_2_-irradiated surfaces than in the control. This might be due to the formation of a new hypercrystalline matrix free from impurities or the presence of non-hydroxyapatite phases, resembling tri- and tetra-calcium phosphates that did not dissolve on exposure to laser [36]. However, there was a significant reduction in these intensities after exposure to acid in comparison to that before pH-cycling associated with lower calcium and phosphorus content (Figure 6, LB, LA). This suggests that CO_2_ laser may not be able to prevent enamel demineralization, but at least it enhanced tissue resistance to dissolution in comparison to untreated surfaces (*p* < 0.000). This might explain the higher microhardness of this group in comparison to all groups before and after pH-cycling (*p* < 0.000), Figure 4. The melting and re-crystallization effects of CO_2_ laser, which occurred at relatively low energy densities (11.5 J/cm^2^), are associated with a decrease in ion release upon exposure to acid due to the melting and sealing effects which are clearly observed in Figure 5 (LB, LA). Chemically, the mineral composition of enamel might be changed to pyrophosphate, tetra-calcium phosphate, or α- and β-tri-calcium phosphate with higher calcium: phosphorus ratio; these changes might contribute to the higher acid resistance of the irradiated enamel surfaces [37]. This is inconsistent with Rodrıguez-Vilchis et al., 2011 [35], Hsu et al., 2000 [36], Liu and Hsu, 2007 [37], and Garma and Jasim, 2015 [38], which is even better than surfaces treated by fluoride [39].

In accordance with the literature [21,40], this study supports the presence of a statistically significant correlation between the chemical composition of enamel and the mechanical gradient of the surface at each scanned point since the microhardness values were increased when the carbonate-to-phosphate peak ratios were decreased (Figure 7). The average ratios of all groups were between 0.02-0.04 before pH-cycling and were increased after pH-cycling in the control and Regenerate-treated groups to 0.33; however, it was lowest in CO_2_ treated-group 0.07. This means that the lower carbonate content in CO_2_ laser-treated surfaces before and after pH-cycling was associated with higher VHN values. Accordingly, the second stated hypothesis was rejected. This is consistent with Freeman et al. (2001) [41] and Crombie et al. (2013) [42], who found that the higher carbonate content reduced the hardness values of enamel. This confirms that the changes in enamel microstructure directly affect its mechanical properties. Nevertheless, the higher carbonate content in the Regenerate group did not suppress the incorporation of phosphate into enamel surfaces making the surface more resistant to acid challenges than in non-treated surfaces.

## 5. Conclusions

Raman microspectroscopy and microindentation detected changes in enamel microstructure and mechanical properties when exposed to acid challenge, in which the intensity of phosphate peaks was significantly reduced post-acid-challenge, but the carbonate was significantly increased. The mineral contents (phosphate and carbonate) were higher in Regenerate-treated surfaces before and after pH-cycling, but the carbonate was the lowest in CO_2_-treated surfaces associated with higher microhardness. Both treatment approaches can efficiently enhance the resistance of enamel and preserve the structure and morphology against acid challenge. 

## Figures and Tables

**Figure 1 diagnostics-13-01087-f001:**
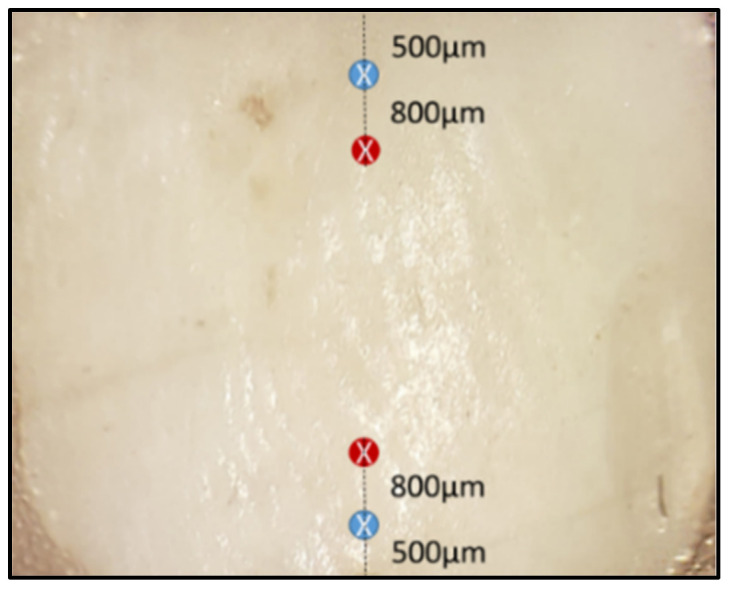
A representative enamel slab illustrates the selected areas that were analyzed by confocal Raman microspectroscopy, followed by Vickers microhardness before pH-cycling (blue circle) and after pH-cycling (red circle).

**Figure 2 diagnostics-13-01087-f002:**
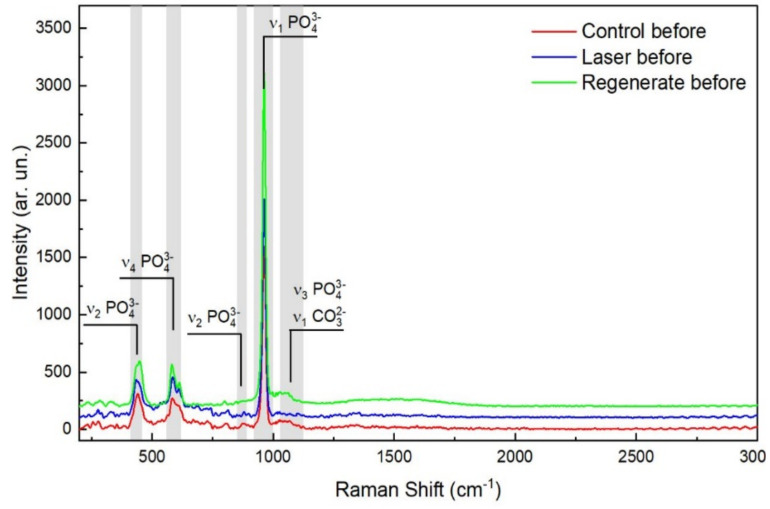
Representative Raman spectra of the control, Regenerate-treated group, and CO_2_ laser-treated group before pH-cycling. The vibrational peak of ν_1_ PO at 960 cm ^−1^ is the strongest signal of all Raman spectra and is the highest in Regenerate-treated group, followed by the laser-treated group, while the control has the lowest peak intensity of all groups. The carbonate at 1070 cm^−1^ is strongest in the Regenerate group.

**Figure 3 diagnostics-13-01087-f003:**
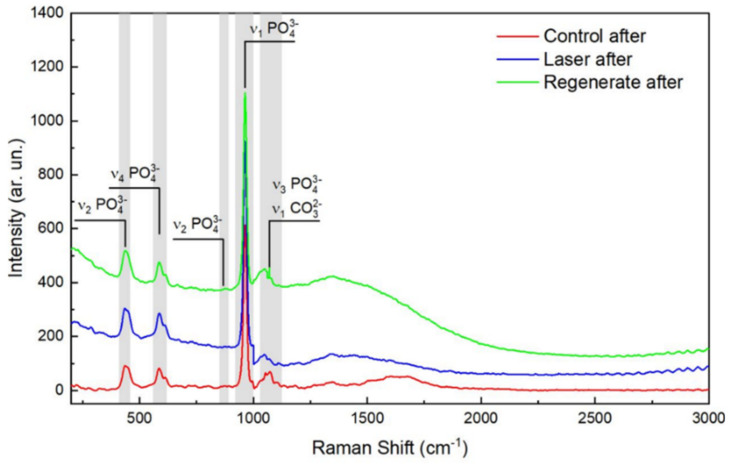
Representative Raman spectra of the control, Regenerate-treated group, and those treated with CO_2_ laser after pH-cycling. The vibrational peak of ν_1_ PO at 960 cm^−1^ is the strongest signal of all Raman spectra and is highest in the Regenerate group, followed by the laser-treated group, while the control group has the lowest peak intensity of all groups. The carbonate at 1070 cm^−1^ is strongest in the regenerate group.

**Figure 4 diagnostics-13-01087-f004:**
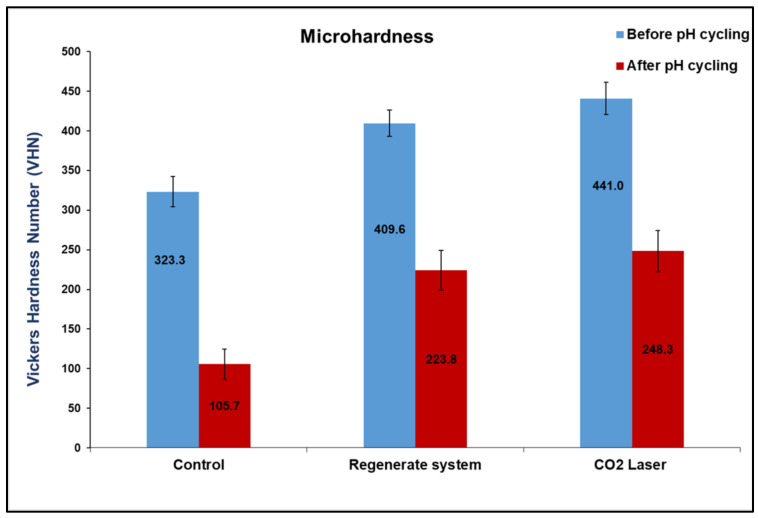
Mean values of Vickers microhardness number (VHN) of the control, Regenerate-treated group, and CO_2_ laser-treated group before and after pH-cycling. The laser-treated group exhibited the highest VH values before and after pH-cycling, followed by Regenerate system (*p* < 0.000).

**Figure 5 diagnostics-13-01087-f005:**
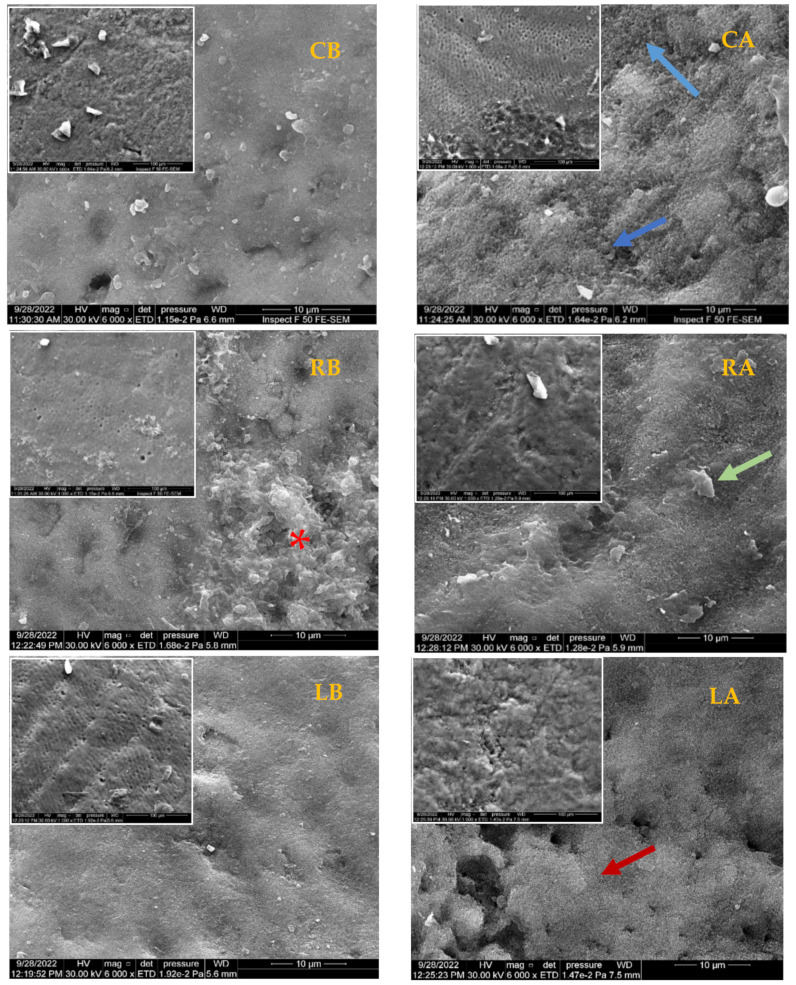
Scanning electron micrograph of representative specimens of control group before and after pH-cycling (CB, CA), Regenerate-treated group (RB, RA), and CO_2_ laser-treated group (LB, LA). The destructive effect of acid challenge is clearly visible in the control group (CA, the blue arrows). The enamel surface that treated by Regenerate appears to be well-covered and protected after acid challenge (RA, the green arrow) with presence of mineral-like deposit at the surface before pH- cycling (RB, red asterisks). In the laser-treated group, the changes in enamel surface (the melting effect) is noted before and after pH-cycling which might make the surface more resistant to the acid attack in comparison to control (LA, the red arrow).

**Figure 6 diagnostics-13-01087-f006:**
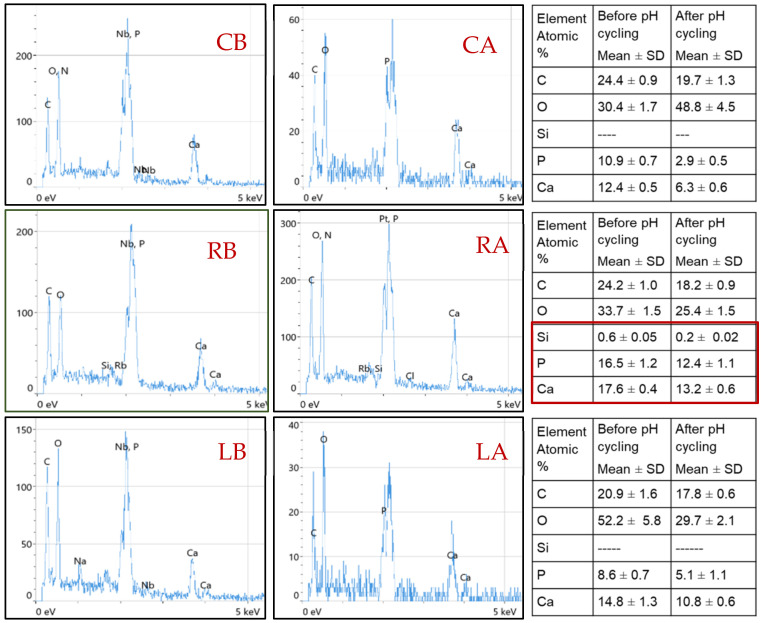
The EDX analysis of the control group (without treatment) before and after pH-cycling (CB, CA), Regenerate-treated surfaces (RB, RA), and CO_2_-treated surfaces (LB, LA). The mineral components (calcium and phosphorus ions) are reduced in all groups after pH-cycling; however, it was higher in Regenerate group (red square) with the presence of silica that is not seen in other groups.

**Figure 7 diagnostics-13-01087-f007:**
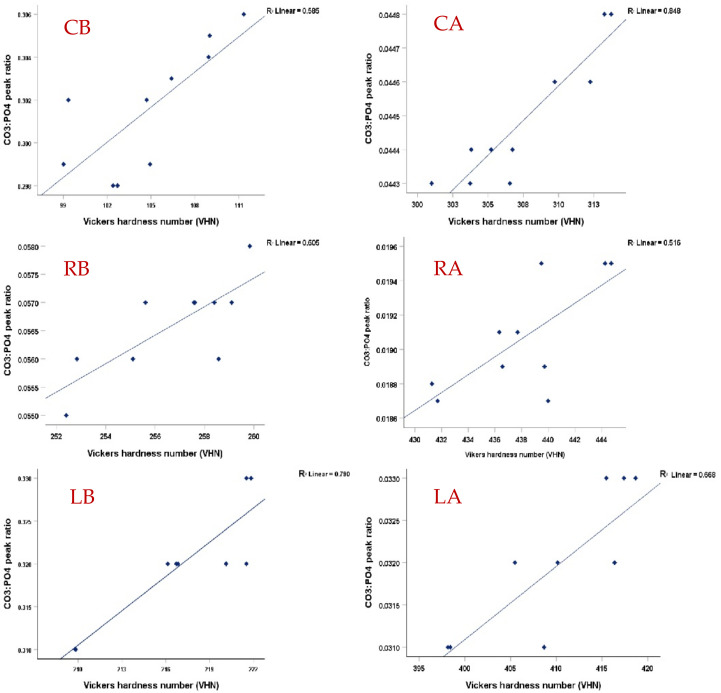
A scatter plot and a regression line (R) demonstrating the presence of a logarithmic regression relationship (*p* < 0.05) between the carbonate to phosphate (1070 cm^−1^/960 cm^−1^) peak ratios and corresponding with Vickers microhardness number (VHN) in each group before and after pH-cycling. The higher peak ratios after pH-cycling in all groups are associated with reduced VHN. The CO_2_-treated surfaces that had lower carbonate content exhibited higher VHN values than other groups before and after pH-cycling (LB, LA).

**Table 1 diagnostics-13-01087-t001:** Raman band intensities A.U. (Mean ± SD) of experimental groups (Control, regenerate system, CO_2_ Laser) before and after pH-cycling.

Raman Peaks	Groups (n = 10)	Peaks Intensities (Mean ± SD) before pH-Cycling	Peaks Intensities (Mean ± SD) after pH-Cycling
Phosphate peaks			
ν_1_-PO_4_ (960 cm^−1^)	Control	1594.99 ± 15.3	779.22 ± 17.4 ^^^
	Regenerate	2811.21± 11.0 *	1072.51± 16.9 *^^^
	CO_2_ Laser	1891.62 ± 16.6 *	940.69 ± 18.0 *^^^
ν_2_-PO_4_ (433 cm^−1^)	Control	260.38 ± 14.9	187.68 ± 11.4 ^^^
	Regenerate	396.54 ± 11.2 *	354.22 ± 10.6 *^^^
	CO_2_ Laser	334.91 ± 11.3 *	247.31 ± 14.4 *^^^
ν_3_-PO_4_ (1029 cm^−1^)	Control	64.56 ± 11.3	182.0 ± 11.0 ^^^
	Regenerate	132.16 ± 11.1 *	344.24 ± 15.9 *^^^
	CO_2_ Laser	40.03 ± 8.1 *	71.94 ± 7.5 *^^^
ν_4_-PO_4_ (579 cm^−1^)	Control	246.21 ± 12.9	158.43 ± 8.3 ^^^
	Regenerate	422.06 ± 14.9 *	355.47 ± 15.8 *^^^
	CO_2_ Laser	352.64 ± 14.5 *	217.31 ± 11.4 *^^^
Carbonate peakCO3 (1070 cm^−1^)	Control	70.66 ± 8.3	242.79 ± 19.7 ^^^
	Regenerate	91.83 ± 6.1 *	355.53 ± 27.2 *^^^
	CO_2_ Laser	36.66 ± 6.7 *	67.76 ± 14.1 *^^^

(*****) Statistically significant differences between the experimental and control groups and between CO_2_ laser and Regenerate (*p* = 0.000). Tukey post-hoc tests (alpha level of 0.05). (^) Statistically significant differences (*p* = 0.000) in each group before and after pH-cycling (Independent *t*-test, *p* < 0.05).

## Data Availability

Not applicable.

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
