# Peer review of "The Effect of a Bioactive Oral System and CO2 Laser on Enamel Susceptibility to Acid Challenge"

_diagnostics, 2023, doi:10.3390/diagnostics13061087_

Round 1

Reviewer 1 Report

This manuscript compared two method (Regenerate Enamel Science System and Carbon Dioxide Laser treatment) of regeneration on the enamel surface after acid changes. However, regeneration method of Regenerate Enamel Science System is not clear scientifically. This means that this manuscript is totally same as product test. So the scientific level of this manuscript is too low. 

Abstract:

L21: What is "VPN"? Abbreviation is needed.

Materials and Methods

Detailed Material components should be added. Especially, Regenerate Enamel Science System is not understandable without any information about composition. This information is very important for scientific approach.

L79: "cemento"→"cement"

L89, L93: similar term "Advanced Tooth Paste system" and "advanced enamel system" ware used.In the title, "Bioactive Oral System" was used also for same system. This is not understandable and it should be unity.

L149,160: Vickers Microhardness→Micro Vickers hardness

L160,165: "Vickers microhardness number (VHN)." and "equivalent micro-hardness values (VHN)" showed same abbreviation. The term shuould be cahnged. 

Conclusion:

L365: "Accordingly, the proposed null hypotheses in this study were rejected." is not needed. This sentence should be removed.

Author Response

Reviewer 1

Comments and Suggestions for Authors

This manuscript compared two methods (Regenerate Enamel Science System and Carbon Dioxide Laser treatment) of regeneration on the enamel surface after acid changes. However, the regeneration method of the Regenerate Enamel Science System is not clear scientifically. This means that this manuscript is totally the same as the product test. So the scientific level of this manuscript is too low. 

Answer: Thank you very much for your comments. This material was previously investigated as a bioactive oral system, the novelty of the current study was to investigate the changes in enamel structure when subjected to pH cycling mimicking the clinical condition and comparing its effect to CO2 laser that was proven to induce chemical changes within the tooth structure making it more resistant to acids. Also, the study explored the presence of a statistical correlation between the carbonate-to-phosphate Raman peak ratio and equivalent Vickers hardness number which is never being investigated for this material.

 Abstract:

L21: What is "VPN"? The abbreviation is needed.

Answer: VHN means Vickers hardness number, amended Line 22

Materials and Methods

Detailed Material components should be added. Especially, the Regenerate Enamel Science System is not understandable without any information about composition. This information is very important for the scientific approach.

Answer: The chemical composition of the system is added Line 116-122

L79: "cemento"→"cement"

Answer: It is a scientific nomenclature that refers to the junction between two different tooth tissues’ layers (enamel and cementum), so it is written in the literature as it “cemento-enamel junction (CEJ) line 102

L89, L93: similar terms "Advanced Tooth Paste system" and "advanced enamel system" were used. The title, "Bioactive Oral System" was used also for the same system. This is not understandable and it should be unity.

Answer: In the title, our opinion was to use the generic group to which the system belongs, and be away from advertising the material, however, if you recommend changing it, we will happy to do. “Regenerate Enamel Science system” is used throughout the text (being highlighted)

L149,160: Vickers Microhardness→Micro Vickers hardness

Answer: it is usually written as one word in the literature

L160,165: "Vickers microhardness number (VHN)." and "equivalent micro-hardness values (VHN)" showed the same abbreviation. The term should be changed. 

Answer: the terminology is changed to “Vickers hardness number (VHN) “ in lines 200-201

Conclusion:

L365: "Accordingly, the proposed null hypotheses in this study were rejected." is not needed. This sentence should be removed.

Answer: Removed in line 506

Finally, we greatly appreciate your efforts in reviewing our work, hoping that we can answer all the questions, wishing that you accept our paper to be considered for publication in Diagnostics 

Best regards from the authors

Reviewer 2 Report

The article is very interesting for the reader.

It is very methodically structured. the section material and methods is well described, like result and discussion. some point have to be discussed:

the introduction can be improved with article concerned the power of CO2 laser , please read these articles:

1.Co2 laser for surgical exposure of impacted palatally canines .

Impellizzeri, A.Palaia, G.Horodynski, M., ...Romeo, U.Galluccio, G. Dental Cadmos2020, 88(2), pp. 122–126 2. 

Histological ex vivo evaluation of peri-incisional thermal effect created by a new-generation CO2 superpulsed laser

Palaia, G.Del Vecchio, A.Impellizzeri, A., ...Russo, C.Romeo, U. The Scientific World Journal2014, 2014, 345685

Author Response

Reviewer 2

Comments and Suggestions for Authors

The article is very interesting for the reader. It is very methodically structured. The section material and methods are well described, like the result and discussion. Some points have to be discussed: The introduction can be improved with article concerned the power of CO2 laser, please read these articles:

  1. Co2 laser for surgical exposure of impacted palatally canines.

Impellizzeri, A., Palaia, G., Horodynski, M.,Romeo, U., Galluccio, G. Dental Cadmos, 2020, 88(2), pp. 122-126 2. 

  1. Palaia, G., Del Vecchio, A., Impellizzeri, A., Russo, C., Romeo, U. Histological ex vivo evaluation of peri-incisional thermal effect created by a new-generation CO2 superpulsed laser The Scientific World Journal, 2014, 2014, 345685.

Answer: Thank you very much for your comments, we greatly appreciate your contribution in reviewing our paper. The introduction was amended for more in-depth critical reviewing using more recent references [Line 42-55, 60-73] 

 Thank you for the papers that you recommend for reading, unfortunately, I uploaded the second one which was added as reference no.14, in lines 60-62 while the other is not accessed, expecting that it was available in the Italian language only. However, the setup parameters for the CO2 laser in our study were adjusted to be used for the preventive effect following the below reference

 Luk, K.; Zhao, I.S.; Yu, O.Y.; Zhang, J.; Gutknecht, N.; Chu, C.H. Effects of 10,600 nm carbon dioxide laser on remineralizing caries: a literature review. Photobiomodul Photomed Laser Surg 2020, 38, 59-65. https://doi.org/10.1089/photob.2019.4690

Finally: we hope that we are able to answer your question, and be considered for publication

Best regards

Reviewer 3 Report

Shubbar et al. examined a series of human premolars treated Regenerate Enamel System and carbon dioxide (CO2) laser to find the structural changes in enamel's morphology and mechanical features. In addition, Raman microspectroscopy and microindentation were utilized to detect changes in enamel microstructure and mechanical properties before and after pH cycling. Unfortunately, the submitted manuscript does not contain novelty or provide any original ideas. Therefore, this study will not add any additional information to this area of research. Nevertheless, I offer some suggestions on how it can be improved. Overall, the manuscript needs serious revision in the places I have indicated before it is suitable for publication.

 1. The introduction needs to be revised. Please provide a more in-depth critical review of previous studies similar to your work, mention what they did and their main findings, and then highlight how your current study brings a new difference to the field.

2. Section 2.4. Chemical nomenclature needs to be standardized.

3. For clarity, the authors should provide more details on Raman spectra preprocessing. For example, it did not mention if the original spectra underwent baseline correction and were normalized or smoothed. It is necessary even if only changes in peak intensity were determined.

4. Because teeth were extracted from patients of different ages, it could introduce some variability in the results. A better solution would be to normalize the spectra and examine changes in the area under the n1 vibration or determine the full width at half maximum (FWHM) at 960 cm-1. Besides mineral content, crystallinity, as expressed by FWHM is another parameter indicating enamel damage or improvement.

5. The intensity of the (PO4)3− ν1 band before and after acid treatment could be used to determine the degree of demineralization.

6. Fig. 2. I am surprised by the presence of a signal from organic material in the spectra of all samples after acid treatment. Are we sure it was still enamel?

7. SEM images were only used to morphologically assess the ability of the products used in this study to protect and remineralize enamel. XRD or AFM methods would be better for assessing the structural alteration of enamel.

8. Fig. 6. The authors should determine the Ca/P ratio with the mean value and SD.

9. Lines 266-268. Please revise the whole statement. Based on the intensity, it is hard to determine a change in the crystallite morphology or orientation of hydroxyapatite crystals that undergo dissolution after acid treatment.

10. Lines 361-362. The mineral contents (phosphate and carbonate) were higher in Regenerate-treated surfaces before and after pH cycling, but the Ca/P ratio was on the same level as the control sample. It needs to be clarified.

11. I recommend introducing some more recent publications to the references.

The weakness of this manuscript is that the discussion reflects speculation rather than the results of actual analysis. Therefore, I recommend major revision along the point of suggestions given above.

I hope these indications and suggestions are able to improve your contribution.

With best regards

Author Response

Reviewer 3

 Comments and Suggestions for Authors

Shubbar et al. examined a series of human premolars treated Regenerate Enamel System and carbon dioxide (CO2) laser to find the structural changes in enamel's morphology and mechanical features. In addition, Raman microspectroscopy and microindentation were utilized to detect changes in enamel microstructure and mechanical properties before and after pH cycling. Unfortunately, the submitted manuscript does not contain novelty or provide any original ideas. Therefore, this study will not add any additional information to this area of research. Nevertheless, I offer some suggestions on how it can be improved. Overall, the manuscript needs serious revision in the places I have indicated before it is suitable for publication.

Answer: Thank you very much for your comments.

Regenerate Enamel Science System is a commercial product that was previously proven to be bioactive since it contains calcium silicate and sodium phosphate salts combined with sodium fluoride. However, the novelty of this work was to figure out whether this material can make a change in the microstructure of enamel and morphology making it more resistant to acid when compared to CO2 laser which was already proven to make such a change, which was never tested before. The study will help the researchers to test any new material or system claimed by the manufacturer for such purpose. Furthermore, the study found a correlation between the chemical (carbonate to phosphate Raman peak ratio (1070/960 cm−1)) and the mechanical properties of enamel (hardness number) at the same points before and after the acid challenge, which will help in the development of a device that can detect these changes clinically rather than in vitro studies only.

  1. The introduction needs to be revised. Please provide a more in-depth critical review of previous studies similar to your work, mention what they did and their main findings, and then highlight how your current study brings a new difference to the field.

Answer: The introduction is amended by adding a more in-depth critical review of the materials that were used in the current study (the highlighted paragraphs, Lines 42-55, 60-73), with adding recent references, however more critical explanations are already present in the discussion section.     

  1. Section 2.4. Chemical nomenclature needs to be standardized.

The answer: thank you for your comment, we run through the chemical nomenclature that was used in the study, and if there is a certain word not written well, can you please mention it 

  1. For clarity, the authors should provide more details on Raman spectra preprocessing. For example, it did not mention if the original spectra underwent baseline correction and were normalized or smoothed. It is necessary even if only changes in peak intensity were determined.

Answer: Thank you for pointing this out. You are correct that preprocessing of Raman spectra is an important step in the analysis of spectral data. In this study, a baseline correction on the original Raman spectra was performed (added in lines 162-163) to remove background noise or systematic fluctuations in the spectra that could affect the accuracy of the analysis, without smoothing or normalization of the spectra. However, the normalization is a useful preprocessing step, but it was not necessary for our analysis, since the focus is specifically on the changes in peak intensity as an indicator of enamel mineral content. Therefore, the analysis of peak intensities was based on the baseline corrected spectra, without smoothing or normalization.

Normalization is not always necessary when analyzing Raman spectra, as its usefulness depends on the research question and the specific properties of the data. In our study, the absolute intensity of the spectra is not important and the focus of the analysis may be on the relative changes in peak position or shape rather than the absolute intensity of the Raman signal. Also, samples are prepared and measured uniformly, so any variations in the intensity of the Raman signal are due to changes in the sample itself rather than external factors such as sample preparation or instrument settings. Finally, if we compare spectra from similar samples with similar Raman acquisition parameters, normalization may not be necessary.

  1. Because teeth were extracted from patients of different ages, it could introduce some variability in the results. A better solution would be to normalize the spectra and examine changes in the area under the n1 vibration or determine the full width at half maximum (FWHM) at 960 cm-1. Besides mineral content, crystallinity, as expressed by FWHM is another parameter indicating enamel damage or improvement.

Answer: Thank you for your comments and suggestions on the analysis of the Raman spectra. You are correct in noting that parameters such as the full width at half maximum (FWHM) at 960 cm-1 can be useful for characterizing changes in enamel crystallinity in addition to mineral content. FWHM is a measure of spectral linewidth that reflects the distribution of frequencies in the Raman spectrum. A narrower FWHM indicates a more ordered or crystalline structure, while a broader FWHM indicates a disordered or amorphous structure. In our study, it is certainly possible that the age range of the teeth included in the study did not involve significant variability in Raman peak width and that no changes in FWHM were observed. Therefore, it may be appropriate to focus on analyzing changes in peak area or intensity, as we did, to assess changes in mineral content.

  1. The intensity of the (PO4)3−ν1band before and after acid treatment could be used to determine the degree of demineralization.

In the result section, this was mentioned in table 1, the highlighted row, in lines (207-209, 213-214, 225-227), and in the discussion section (Lines 406-412)

  1. Fig. 2. I am surprised by the presence of a signal from organic material in the spectra of all samples after acid treatment. Are we sure it was still enamel?

Answer: The presence of a small fluorescence signal in the Raman spectra acquired from enamel samples after acid treatment is expected, as it is well known that acid treatment can cause partial demineralization of enamel, which can lead to the exposure of organic material, such as proteins and lipids. These organic materials can contribute to the observed fluorescence signal in the Raman spectra. In addition, acid treatment can change the crystal structure of the enamel, which can affect the fluorescence properties of the sample. For example, acid treatment can alter the distribution of mineral and organic components within the enamel, leading to changes in the local environment and potentially increasing the fluorescence signal. However, it is important to note that the fluorescence signal observed in the spectra is relatively small and is consistent with the expected amount of organic material that would be present in enamel samples, even after acid treatment. The presence of this signal is not necessarily an indication that the samples are no longer enamel.

  1. SEM images were only used to morphologically assess the ability of the products used in this study to protect and remineralize enamel. XRD or AFM methods would be better for assessing the structural alteration of enamel.

Answer: Thank you very much for your comments. While XRD and AFM techniques can provide valuable information on the structural changes of enamel, I would respectfully disagree with the suggestion that SEM is insufficient for the study described. SEM is a powerful tool for studying enamel morphology and topography at high resolution. It allows direct visualization of enamel surfaces and can provide information on the size, shape, and distribution of enamel crystals as well as surface changes or defects. Unfortunately, the XRD technique is not well suited for the analysis of small samples (4.0 × 4.0 × 2.0 mm), so it may not be used in studies with small or limited samples, as is the case in our study.

  1. Fig. 6. The authors should determine the Ca/P ratio with the mean value and SD.

Answer: Each number in the table is already representing the mean, the SD was added to the values in Fig 6, however, the Ca/P ratio was not calculated in the study, since the EDX was used to determine the amount of mineral loss from enamel after pH-cycling among groups to evaluate the preventive effect of Regenerate or CO2 in comparison to the control (intact surfaces)

  1. Lines 266-268. Please revise the whole statement. Based on the intensity, it is hard to determine a change in the crystallite morphology or orientation of hydroxyapatite crystals that undergo dissolution after acid treatment.

Answer: This sentence was amended [There was a considerable reduction in the intensity of phosphate peaks in all groups after pH-cycling (p<0.000, Table 1), which might indicate the dissolution of the hydroxyapatite (HAp) crystals in enamel when subjected to an acidic condition]

  1. Lines 361-362. The mineral contents (phosphate and carbonate) were higher in Regenerate-treated surfaces before and after pH cycling, but the Ca/P ratio was on the same level as the control sample. It needs to be clarified.

Answer: This study assessed the structural changes in enamel based on changes in the intensity of phosphate peaks using Raman spectroscopy as suggested in our hypothesis, however, further analysis using EDX was used to support the loss of ions (P, Ca, and Si) after pH cycling among groups [ This was mentioned in Line 413-417]

Whilst, assessing the Ca/P ratio in these surfaces via EDX was not mentioned in our hypothesis, since the Raman was already used to identify the changes in the HAp based on the intensity of v1-v4, and from the scientific background is considered more specific for such a purpose. The calculation of the Ca/P ratio based on EDX is not valid in our study since it was increased after pH cycling in the control group, which is really need to be further investigated by using another specific analytic device such as XRD to support or deny that the EDX is a good tool to measure the Ca:P ratio.  

  1. I recommend introducing some more recent publications to the references.

Answer: more new references were added (The highlighted references)  

The weakness of this manuscript is that the discussion reflects speculation rather than the results of actual analysis. Therefore, I recommend major revision along the point of suggestions given above.

Answer: There is a clear description of the result in the discussion correlated to the hypotheses that were stated in our study, and the explanation for these results either proved in our work (SEM-EDX and microhardness) or based on previous studies.

Finally: we greatly thank you for your respected efforts in reviewing our work, hoping that we are able to answer your questions clearly and amend the manuscript to be better, wishing that it will be considered for publication  

Best regards

Round 2

Reviewer 1 Report

The revised manuscript looks improved. However, still I have minor concerns. 

L22: Vickers hardness number (VHN)→Vickers hardness (VH)

L200: Vickers hardness number (VHN)→VH

Author Response

Thank you for your comments, we are absolutely happy if you mention how we can improve our manuscript, as it was designed to assess the chemical, morphological, and hardness properties of enamel after applying two systems as a preventive measure to improve the resistance of enamel to an acidic challenge. The hypothesis that was proposed in our study was followed by careful analysis of data with the aid of a statistician. The findings are clearly discussed in the discussion section, while the conclusion supports the use of both techniques for the claimed purpose. If you have further suggestions to improve the quality of the manuscript following the aim of the study and the hypothesis that was mentioned, we will be happy to do further amendments. 

Answer: L22: Vickers hardness number (VHN)→higher microhardness values

Line 200: Vickers hardness number (VHN), remains as it is, since the statistical analysis was performed to find a correlation between Raman phosphate: carbonate peak ratio and their equivalent Vickers hardness number (VHN) at each point. 

I wish that is clear now

Please accept our best regards

Reviewer 3 Report

I consider that the authors have improved the article as much as possible.

Author Response

Thank you for your comments

If you have further suggestions to improve the quality of the manuscript following the aim of the study and the hypothesis, we are glad to do so

Please accept our best regards